# Polyunsaturated Fatty Acids: Conversion to Lipid Mediators, Roles in Inflammatory Diseases and Dietary Sources

**DOI:** 10.3390/ijms24108838

**Published:** 2023-05-16

**Authors:** John L. Harwood

**Affiliations:** School of Biosciences, Cardiff University, Cardiff CF10 3AX, Wales, UK; Harwood@cardiff.ac.uk; Tel.: +44-2920-874108; Fax: +44-2920-874116

**Keywords:** polyunsaturated fatty acids, synthesis, lipid mediators, inflammation, eicosanoids, docosanoids, specialised pro-resolving mediators

## Abstract

Polyunsaturated fatty acids (PUFAs) are important components of the diet of mammals. Their role was first established when the essential fatty acids (EFAs) linoleic acid and α-linolenic acid were discovered nearly a century ago. However, most of the biochemical and physiological actions of PUFAs rely on their conversion to 20C or 22C acids and subsequent metabolism to lipid mediators. As a generalisation, lipid mediators formed from n-6 PUFAs are pro-inflammatory while those from n-3 PUFAs are anti-inflammatory or neutral. Apart from the actions of the classic eicosanoids or docosanoids, many newly discovered compounds are described as Specialised Pro-resolving Mediators (SPMs) which have been proposed to have a role in resolving inflammatory conditions such as infections and preventing them from becoming chronic. In addition, a large group of molecules, termed isoprostanes, can be generated by free radical reactions and these too have powerful properties towards inflammation. The ultimate source of n-3 and n-6 PUFAs are photosynthetic organisms which contain Δ-12 and Δ-15 desaturases, which are almost exclusively absent from animals. Moreover, the EFAs consumed from plant food are in competition with each other for conversion to lipid mediators. Thus, the relative amounts of n-3 and n-6 PUFAs in the diet are important. Furthermore, the conversion of the EFAs to 20C and 22C PUFAs in mammals is rather poor. Thus, there has been much interest recently in the use of algae, many of which make substantial quantities of long-chain PUFAs or in manipulating oil crops to make such acids. This is especially important because fish oils, which are their main source in human diets, are becoming limited. In this review, the metabolic conversion of PUFAs into different lipid mediators is described. Then, the biological roles and molecular mechanisms of such mediators in inflammatory diseases are outlined. Finally, natural sources of PUFAs (including 20 or 22 carbon compounds) are detailed, as well as recent efforts to increase their production.

## 1. Introduction

It was the husband and wife team of George and Mildred Burr which demonstrated for the first time that lipids were a vital component of a healthy diet for mammals [1]. A year later, they showed that the polyunsaturated fatty acids (PUFA) linoleic acid [2] and later α-linolenic acid [3] could alleviate the symptoms produced by a fat-free diet. Although controversial at the time, there were, thus, two distinct types of PUFAs which were both essential—the n-3 (omega-3) and the n-6 (omega-6) families [4]. In their experiments (using fat-free diets) a variety of physiological symptoms were observed which were characteristic of essential fatty acid (EFA) deficiency and these characteristic features could be abrogated by feeding linoleic and α-linolenic acids [4]. These acids can be regarded as key to preventing EFA deficiency and, as we will see later, are essential in a normal healthy diet [5].

While only small amounts of linoleic and α-linolenic acids are needed to prevent EFA deficiency, it has become increasingly recognised that the n-3 and n-6 PUFAs have further dietary importance, mainly because they can be metabolised to a whole series of lipid mediators. The latter compounds are usually formed from longer-chain (20C or 22C) PUFAs, and there has been increasing interest in such acids as dietary components. Research has primarily centred on arachidonic (ARA), n-3 eicosapentaenoic (EPA) and docosahexaenoic (DHA) acids. Moreover, because ARA mainly gives rise to pro-inflammatory or inflammatory mediators whereas the n-3 PUFA products have the opposite effect, attention has also been paid to the n-6/n-3 ratio in diets with the advice this should be about 3–4, which is much lower than current ‘Western’ diets [6].

Many of the most prevalent diseases have chronic inflammation as an important determinant [7]. Thus, it is not surprising that there has been much interest in the role that dietary PUFAs could play in increasing the risk of onset or in ameliorating such diseases through the production of useful lipid mediators [8]. Thus, for example, a potential therapeutic target for atherosclerosis has been suggested for mediators that reduce NFκB inflammatory effects and resolve inflammatory episodes [9,10]. By the same token, it has to be recognised that both n-3 and n-6 PUFAs are needed for good health. Table 1 lists diseases where n-3 PUFAs have been shown to have beneficial effects. The major complaints will be dealt with in more detail in later sections.

Uncontrolled and/or chronic inflammation is important in many widespread pathological situations. These include arthritis, cancers, cardiovascular diseases, chronic pain, and neurological complaints, as well as bacterial and viral (including COVID-19) infections [11,12]. Thus, there is an urgent need to understand the mechanism(s) of inflammation so that new treatments can be developed [13,14]. Current treatments, such as the use of steroids or non-steroid anti-inflammatory drugs (NSAIDs) [15] are clearly inadequate and, indeed, have unwanted side effects. Therefore, the development of new concepts and novel mediators is urgently required [11,16,17].

## 2. Conversion of Polyunsaturated Fatty Acids into Lipid Mediators

The production of lipid mediators from PUFAs can be considered in two main sections. First, there is the conversion of EFAs into classic eicosanoids. This aspect has been covered in some detail recently [18,19,20] and, in particular, the molecular mechanisms of the three types of oxidases involved have been described in detail [21]. Second, there is the production of specialised pro-resolving mediators (SPMs) which can be formed by both enzymatic and non-enzymatic PUFA oxidation [11,22].

### 2.1. Eicosanoid Biosynthesis

The biosynthesis of the classic eicosanoids (20C oxidised derivatives of PUFAs) is due to the action of three different types of oxidation. The enzymes involved are the cyclooxygenases, lipoxygenases, or the cytochrome P450 oxidase/epoxygenases [18,19,21,23]. All these reactions utilise a non-esterified PUFA which is usually released from a membrane phosphoglyceride by phospholipase A2 (PLA2) action. A number of PLA2 enzymes have been characterised [24]. These can be categorised into cytosolic Ca-dependent (cPLA2), cytosolic Ca-independent (iPLA2) and secreted PLA2 (sPLA2) groups. Moreover, the main substrate fatty acids (ARA, EPA, DHA) have different phospholipid sources [18] and, to an extent, different activities with the various hydrolytic enzymes [25]. Further details of the physiological interactions that influence the PLA2-mediated release of PUFAs from membrane lipids may be found in [18,19].

Oxidation of the 20C PUFA (usually ARA or EPA) can be through cyclooxidase action [26]. There are two major human cyclooxygenase isoforms (COX-1, COX-2) which differ in their expression and substrate selectivity [18]. COX-1 is expressed constitutively in most mammalian tissues while COX-2 has a characteristic low expression unless induced by inflammatory stimuli such as cytokines or tumour promoters [27,28]. The main exceptions to this are that both isoforms (COX-1, COX-2) are expressed constitutively in the brain, testes, and the *macular densa* of the kidney [29]. Two reactions are catalysed by both COXs. In the first, two molecules of oxygen are added to the substrate by a cyclooxygenase reaction. For ARA this leads to prostaglandin PGG2. The latter intermediate is then reduced during a peroxidase cycle to PGH2. The COX enzymes are more correctly called prostaglandin endoperoxide H synthase. Although COX-2 has a wider substrate selectivity (including n-3 PUFAs) than COX-1 [27,28], ARA is still the preferred substrate for both [30].

Once the endoperoxide intermediate has been produced by COX activity, it can be converted into three classes of prostanoids: prostaglandins (e.g., PGD2, PGE2, PGF2), prostacyclin (PGI2), or thromboxane TXA2—all from PGH2 (Figure 1). PGI2 promotes vasodilation, inhibits platelet aggregation, and contributes to myocardial protection. In contrast, TXA2 promotes platelet aggregation [31]. Since these two prostanoids have antagonistic effects, their balance is important for vascular homeostasis and healthy cardiovascular function [32]. Notably, prostanoid synthases have isoforms with variable distributions in different tissues. Thus, the eicosanoid composition produced in individual tissues can vary significantly [19].

The isolation, purification, and characteristics of enzymes involved in the production of prostanoids via COX activity are detailed in [33]. In addition, details of the structure of COXs as well as their catalytic mechanisms and substrate selectivities are described in [21]. COX-1 and possibly COX-2 can undergo an irreversible substrate-independent self-inactivating process [34], possibly due to radical intermediates formed in either of the two partial reactions [33].

All the lipid mediators described here have local and normally transient actions in tissues. For prostanoids, they can be metabolized to other less-active metabolites (Figure 1) or, in the case of 15d-prostaglandin J2, they can bind to PPAR-γ [35]. In turn, such binding can have anti-inflammatory and anti-oxidant actions e.g., via inhibition of NFκB and JAK-STAT pathways [36]. For many prostaglandins, 15-PGDH (15-hydroxyprostaglandin dehydrogenase) is a key enzyme. It causes the rapid oxidation of 15-hydroxy-prostaglandins into their less active 15-keto derivatives. It is ubiquitously expressed in mammalian tissues and is particularly important for PGD2, PGE2, and PGF2α catabolism.

Lipoxygenases (LOXs) can catalyse three different types of reactions: hydroperoxide production via dioxygenation, hydroperoxidation which yields keto lipids, and a leukotriene synthase reaction to give epoxy leukotrienes [18]. They have also been proposed to be involved in the biosynthesis of SPMs. In humans, six main families of LOXs have been identified—5-LOX, 12-LOX, 12/15-LOX (15-LOX type 1), 15-LOX type 2, 12*R*-LOX and epidermal LOX [37]. Traditionally, LOXs have been named according to their positional specificity for arachidonic acid so that, for example, 15-LOX would be called ALOX-15 [21]. In mice, there are seven LOXs. All of the animal LOXs have a single polypeptide chain of 75–80 kDa mass. A beta-barrel at the N end is used for substrate acquisition and a catalytic domain contains a single non-heme iron atom. Other features of the reaction, such as the role of iron and molecular interactions of the substrate with the lipoxygenase, are summarised in [18]. In humans, there are six functional genes (ALOX5, ALOX12, ALOX12B, ALOX15, ALOX15B, ALOXE3), four pseudogenes, and an ALOX12 antisense gene [38].

5-LOX is involved in the formation of leukotrienes (Figure 2) and is expressed mainly in cells of myeloid origin. In resting cells, 5-LOX is a soluble enzyme in the cytosol. After cell activation, it co-migrates with PLA2 to the endoplasmic reticulum [19]. It can also interact with cytosolic phospholipase A2 and can be phosphorylated at multiple sites [21]. Moreover, after phosphorylation, 5-LOX can translocate to the nuclear envelope [39]. 5-LOX catalyses a two-step concerted reaction with two accessory proteins (5-lipoxygenase activating protein (FLAP), coactosin-like protein (CLP)) being needed [21]. The product from ARA is LTA4 and this can then form LTB4 or, alternatively, the cysteinyl leukotrienes (Figure 2).

LOXs are usually constitutively expressed except for 15-LOX (ALOX15) which can be induced by IL-4 or IL-13 in macrophages [40]. Studies on mammalian LOXs show that the N-terminal domain can function for membrane-binding and regulation [41,42], such as in the translocation from the cytosol to the cell membrane [21].

Crystal structures are available for the 5- and 15-LOXs which show a single polypeptide chain with the characteristics outlined previously. Other crystal structures for various LOXs are noted in [43]. The coordination positions of the catalytic iron are occupied by three conserved histidine residues, the carboxyl group of the C-terminus Ile, and a variable ligand [43]. Although all human LOXs were thought to function as monomers, studies with the human platelet 12S-LOX (ALOX12) revealed it as a dimer [44]. Indeed, human 5-LOX can function as a monomer or a dimer [45]. The reaction mechanism for the dioxygenase activity of LOX and a note of other reactions involving free-radical processes are discussed well in detail by Hajeyah et al. [21]. Some LOXs exhibit suicide inactivation. For example, human 5-LOX is irreversibly inactivated by both 5-HpETE and leukotriene A4 products [46]. Further molecular mechanisms for catalysis and regio [47,48] and stereoselectivities [49] are discussed in [50].

Instead of being converted to leukotrienes, ARA can form lipoxins, which are considered a class of SPMs. The biosynthesis of lipoxin (5,6,15-trihydroxy-7,9,11,13-eicosatetraenoic acid) (LX) is carried out by interactions between different cells. The process involves three lipoxygenases (5-,12-, and 15-LOX) and, in addition, aspirin-triggered lipoxins (ATL) can be formed [51]. All the LXs are strongly anti-inflammatory [52]. The aspirin-triggered lipoxins are 15-isomers of LX [53] but have been reported to be more resistant to metabolic inactivation [54]. Two other products of LOX action are eoxins (which are strongly pro-inflammatory) [55] and hepoxilins. The latter are important in human epidermis function [56].

The third type of oxidation that can occur on PUFAs such as ARA is catalysed by the cytochrome P450 oxidases (CYPs). They belong to a superfamily of heme-containing mixed-function monooxygenases [57]. In humans, there are over 50 functional genes and their active enzymes are widely distributed with important functions in health and disease [58]. The CYPs transfer a single oxygen to the substrate and are membrane-bound haem proteins. NADPH is needed and comes via an NADPH-cytochrome P450 reductase. They produce a mixture of hydroxy-eicosatetraenes (HETEs) due to their omega-hydroxylase activities, the mixture depending on the individual isoforms [59]. Alternatively, epoxygenase activity allows the production of cis-epoxyeicosatrienoic acids (EETs) which have major functions as autocrine and paracrine effectors in the renal and cardiovascular systems. EETs are rapidly metabolised to their corresponding dihydroxyeicosatrienoic acids (DHETs) which have significantly smaller physiological activities [18].

As is the case with COX and LOX products, research on products of cytochrome P450 oxidases has concentrated on metabolites of ARA [59]. However, it should be noted that the latter oxidases often have a preferential selectivity towards n-3 PUFAs such as EPA or DHA [18]. Furthermore, oxidation of linoleic acid can take place to form octadecanoids which can influence the inflammation associated with metabolic syndrome and cancer [60,61].

The oxylipins produced by CYPs are dependent upon and easily altered by diet [62,63]. CYPs primarily convert linoleic acid into epoxyoctadecamonoenoic acid (EpOME) while α-linolenic acid (ALA) is converted mainly into epoxyoctadecadienoic acid (EpODE) [63]. Since linoleic acid is the main PUFA in human diets it is important to note that CYP products tend to be pro-inflammatory. They can act by activating NFκB to give oxidative stress [64,65] or to change lipid metabolism through PPARγ [66]. For the ALA-derived oxylipins, much less is known but some products inactivate the NLRP3 inflammasome [66] suggesting that they may play a role in reducing inflammation [62].

Our current knowledge of the structure of mammalian CYPs is described in [21]. There are two domains; a beta-sheet-rich N-terminal domain and a larger helix-rich C-terminal catalytic region. A transmembrane helix is located in the N-terminal domain. Important structural features and the general molecular reaction mechanism are described in [21,67].

Apart from the 18C oxylipins, CYP products of ARA [57] have been shown to have important effects on inflammation. EETs reduced cytokine-induced endothelial cell adhesion molecule expression and leukocyte adhesion to the vascular wall by inhibiting transcription factor NFκB [68]. Moreover, later work showed that 14,15-DHET,11,12-EET and 20-HETE were all potent activators of PPARα [69]. PPARα can then inhibit the inflammatory response by repressing NFκB signalling. Expression of several genes involved in inflammation, including COX-2, was inhibited after PPARα activation [70].

### 2.2. Eicosanoid Receptors

It has become apparent over the last two decades that most eicosanoids mediate their effects through G-protein-coupled receptors [71,72,73]. A list of G-protein-coupled eicosanoid receptors along with their ligands and signalling systems is provided in [19]. In addition, a comprehensive recent review of prostanoid receptors included details of their expression, characterisation, regulation and mechanism of action [74]. Although each receptor binds with highest affinity to a particular prostanoid, there is considerable cross-reactivity towards different prostanoids. Thus, receptors DP1 and DP2, whose affinities are highest for PGD2, will also bind PGE2 and PGF2α. Moreover, some prostanoids can bind to several receptors. For example, PGF2α can bind to both DP receptors and all four EP receptors, as well as those for prostacyclin and thromboxane. Moreover, the 1-series prostaglandin, prostacyclin, can bind to the prostanoid receptor IP but also to PPARδ [75]. Furthermore, the EETs can bind to PPARα and PPARγ and, in addition, may have their own G-protein-coupled receptors [19].

Prostaglandin D2 can be bound to either DP1 or DP2 which are distinct receptors [76]. Both receptors are coupled to G-proteins and, upon binding by PGD2 (or its PGJ2 series products) will change intracellular cAMP and Ca++ concentrations [74]. These alterations for DP1 do not involve inositol-1,4,5-trisphosphate [77]. As with most G-coupled receptors, regulation of DP1 and DP2 receptors is accomplished by desensitisation and internalization but with some detailed differences. Desensitization begins with phosphorylation of the receptor leading to the high-affinity binding of arrestins [78]. An example of the differences between DPI and DP2 is where activation of the latter changes Ca++ concentrations via both PI3K and Ca++/calmodulin/calcineurin signalling [79]. The synergy and opposing action of DP1 and DP2 receptors are detailed in [74] where further descriptions of prostaglandin E2 (EPs), prostacyclin (IP), thromboxane (TP) and prostaglandin F2α (FP) receptors will be found.

Fujino [80] has recently proposed a new viewpoint for G-protein-coupled receptor-mediated signalling. Instead of a single pathway being activated from a receptor, the new concept allows more than one pathway to be activated from a single receptor. Thus, prostanoids can act as ‘biased’ ligands and provide multiple benefits or disadvantages.

## 3. Specialised Pro-Resolving Mediators (SPMs)

Recently, a multi-author publication [81] has questioned some aspects of the formation and function of SPMs. This is a controversial area which is discussed in detail at the end of Section 6.3.3. Because the disagreements have not yet been resolved, it should be mentioned at this point that they have implications for Section 3 and Section 6.

Specialised pro-resolving mediators are so called because of their proposed function in the resolution stage of inflammation and the first representative was isolated some 20 years ago [82]. Inflammation itself is a natural protective process where the body responds to harmful stimuli such as pathogens or wounding. It functions to clear out damaged tissues and necrotic cells as well as eliminate the initial cause. Acute inflammation is actively terminated by its resolution which has been proposed to involve several mechanisms including the production of SPMs, as well as the down-regulation of pro-inflammatory substances (such as leukotrienes). If resolution does not occur, chronic inflammation will ensue with severe consequences [11,22].

As mentioned above, the first SPM which was isolated was a resolvin formed from EPA [82]. Four individual resolvins (E-series resolvins e.g., RvE1) can be formed from EPA and have been characterised [83]. Six D-series resolvins are formed from DHA (RvD1, RvD2 etc.) [84] while n-3 docosapentaenoic acid can form D-series resolvins (e.g., RvDn-3DPA) as well as the 13-series resolvins (RvTs) [11,22]. Some details of the enzymatic reactions and molecular details for the generation of E-resolvins via COX-2 and CYPs have been discussed in [21]. 18*R*-HEPE can be converted by 5-LOX into a 5*S*-hydroperoxy intermediate and then to a 5*S*,6*S*-epoxyintermediate [85]. The hydroperoxyl intermediate can be peroxidised to resolvin E2 while the epoxy intermediate can be hydrolysed by LTA4H to resolvin E1 [86]. 18S-HEPE has also been proposed to form 18*S*-analogues of resolvin E1 and E2 [21].

It is, perhaps, ironic that aspirin (which is used widely along with other NSAIDs to inhibit COX activity) [87] should have proven useful in the discovery of many SPMs. Thus, aspirin treatment can lead to the formation of 17*R*-resolvins from DHA as well as the aspirin-triggered resolvins AT-RvD1 and AT-RvD2 [18]. The discovery of these reactions [82] and further details are given in [88].

As an alternative to resolvins, DHA has been proposed to be modified by 15-LOX to generate protectins or by 12-LOX to form maresins. The first protectin (NPD1) was isolated from the brain and was termed neuroprotection because of this and its protective action [84]. However, because similar docosanoids have now been reported in many tissues, the term ‘protectin’ is now preferred for their other peripheral actions [11,18]. Aspirin irreversibly inhibits COX-1 at its active site but only partly blocks that of COX-2 and modifies its activity. The latter can show activity similar to 15-LOX but with a difference in the configuration for oxygen insertion [18]. The aspirin-triggered epimer, 17*R*-NPD1 shares the actions of NPD1 by attenuating stroke, controlling polymorphonuclear leukocytes and facilitating macrophage activity [11].

A recent review has provided more details about the biosynthesis, catabolism, and structure-functions of the protectins [89]. Particular emphasis has been placed on structure-function data for sulfido-conjugated protectins as well as the catabolism of protectin D1. It was pointed out that by 2022 some 43 SPMs possessing pro-resolution and anti-inflammatory reactions had been reported. Because mediators such as PD1 act locally, they then have to be catabolized rapidly. The role of β-oxidation in this process has been detailed [90].

Maresin synthesis has been proposed to begin with 12*S*-LOX generating 14*S*-hydroperoxy-DHA, followed by an epoxy-intermediate. The epoxy-intermediate can then be hydrolysed by different enzymes [21] to generate maresins 1 or 2 [91,92].

Recently, three series of peptide lipid-conjugated SPMs have been reported. These are conjugates with maresins, protectins, or resolvins and have been termed cys-SPMs [93]. Each series has three members, all with potent pro-regenerative and pro-repair actions [11]. Data from detailed structure-function analyses of these sulfido protectin conjugates as well as some protectin analogues have been published [89].

In contrast to the SPMs formed from n-3 PUFAs, ARA can be converted to lipoxins, which are trihydroxyeicosatetraenoic acids [39,94]. There are at least three routes for the biosynthesis of lipoxins. LXA4 and LXB4 can be generated by 5-LOX action on 15*S*-HPETE [95]. Alternatively, a transcellular pathway using 5-LOX in one cell and 12-LOX in another has been reported [96]. A third mechanism produces epi-lipoxins (or aspirin-triggered lipoxins; ATL) using an aspirin-modified COX-2 which forms 15R-HPETE and then 5-LOX continues the process [95]. Catabolism of the lipoxins occurs via 15-hydroperoxyprostaglandin dehydrogenase and prostaglandin reductase. The ATL share the anti-inflammatory actions of AXs but are more resistant to metabolic inactivation [54]

A complex interaction between receptors and enzymes has been reported for lipoxin generation in macrophages [97]. Lipoxins were generated following sequential activation of toll-like receptor 4 (TLR4, a receptor for endotoxin) and P2X7. Initial activation of TLR4 led to an accumulation of 15-HETE within membrane phospholipids. Activation of P2X7 caused the release of 15-HETE through cPLA2 action. This allowed 5-LOX to produce enhanced lipoxin levels.

### 3.1. Receptors for SPMs

As discussed in [18], lipid mediators bind to receptors in order to facilitate their cellular actions. Apart from those for prostanoids and leukotrienes discussed previously, receptors have been reported for lipoxins [98] and other SPMs [99] (see Figure 3).

In summary, each SPM has been suggested to promote resolution by interacting with a specific G-protein-coupled receptor (GPCR) in a stereospecific manner [11]. The affinities are reported to be in the low nM range which is consistent with their in vitro and, in particular, in vivo biological activities. Details of the individual-reported GPCRs are given in [11,22]. This includes recently discovered receptors such as GPCR37 for NPD1/PD1 [100], LGR6 for maresin MaR1 [101], and GPR101 for RvDRn-3DPA [102]. Although there are a considerable number of SPMs reacting with specific receptors, interactions can often produce similar cellular changes such as alterations in cAMP or Ca^++^. Thus, receptor activation by different SPMs can initiate overlapping functions [11] but see also the detailed comments in [81] about the receptors.

The receptor for lipoxin A and formyl receptor 2 is ALX/FPR2. It possesses high sequence homology to the formyl peptide receptors [103]. It will also ligate ATL [104], resolving D1 [105] and D3 [106]. Genetic targeting of ALX/FPR2 in different murine models has generated conflicting results. This is possible due to variable balances between pro-inflammatory and pro-resolving agonists in the various models [107]. LXA4 is also a ligand for the nuclear aryl hydrocarbon receptor [108].

### 3.2. Further Metabolism of SPMs

Just as prostanoids have their effects locally and are rapidly catabolised, so the SPMs are quickly metabolised through enzyme inactivation [11]. Oxidative enzymes are active, and for RvE1 at least four distinct pathways are involved [18]. As will be seen in Figure 3, both cytochrome P450 and, especially, 15-hydroxyprostaglandin dehydrogenase have been reported to be active. Because of the proposed beneficial actions of SPMs during inflammation, metabolically stable analogues (that resist inactivation) have been created [109]. In particular, benzo-diacetylenic derivatives and several mimetics for RvD1 have been identified. The creation of a number of different structures as mimetics should allow them to be used as templates in the clinical development of useful therapeutics for reducing inflammation in vivo [11].

## 4. Auto-Oxidation of PUFAs to Create Anti-Inflammatory Products

A major group of compounds was discovered much later than the eicosanoids when the auto-oxidation of ARA by reactive oxygen species (ROS) through free radical reactions was characterised [110]. These compounds are termed isoprostanes, of which phytoprostanes are produced by ROS action on ALA in plants and neuroprostanes are formed by free radical-mediated peroxidation of DHA [111]. As noted above, the earliest studies on isoprostanes (IsoPs) were with ARA and, more recently, this PUFA has continued to be well studied. IsoPs can be considered to be geometric isomers of prostaglandins but where their lateral chains are in the *cis* configuration (rather than the *trans* configuration). IsoPs exert their effects through the activation of prostanoid receptors or via kinase receptors (e.g., tyrosine, Rho) [111]. In the main, IsoPs are formed by free radical reactions on PUFAs esterified to phosphoglycerides rather than using unesterified fatty acids, as for the prostanoids. The oxidation reactions produce a mixture of racemic IsoPs in equal amounts. They are released from phosphoglycerides by phospholipase A2 or platelet-activating factor hydrolase [112] after which further modifications can take place. IsoPs influence numerous diseases and are important in hypertension and ischemic health. Their metabolism and function have been covered excellently in several recent reviews to which the reader is referred [22,111,112,113,114] for further details.

## 5. Fatty Acid Esters of Fatty Acids (FAHFAs)

Hydroxylated fatty acids esterified to another fatty acid to form FAHFAs have been found in a host of different tissues. Following their first identification in white adipose tissue, they have been found in serum, milk and many other tissues from mice and humans. In addition, FAHFAs are present in insects, plants and bacteria. Moreover, other lipids containing hydroxy fatty acids are present in a variety of forms and in many different tissues and organisms [115].

Although the complete biosynthetic pathway for FAHFA formation in mammals is still unclear [115], it is known that they are formed endogenously with defined stereochemistry. Thus, the hydroxyl is in the R-configuration and peroxisomal enzymes are believed to be involved. The adipose tissue triacylglycerol stores are a source of FAHFAs. More is known about the biosynthesis of some plant FAHFAs [115].

FAHFAs have been reported to have beneficial biological effects in a variety of mammalian tissues including adipose tissue, gut, pancreas, liver, and muscles. Anti-inflammatory actions have been reported after lipopolysaccharide-stimulated dendritic cell and macrophage activation [116] as well as modulation of innate and adaptive immune responses for type-1 diabetes [117] and in a mouse colitis model [118]. Actions mediated through GPR120 and GPR40 receptors have been reported for these anti-inflammatory effects. For further details on the chemical synthesis of FAHFAs, the diversity of different FAHFA families, and their metabolism and biological effects, see the comprehensive review [115].

## 6. Molecular Mechanisms and Biological Effects of Lipid Mediators

### 6.1. Role of Eicosanoids in Inflammation and Immunity

The various eicosanoids (prostaglandins, thromboxanes, prostacyclin, leukotrienes, HETEs, EETs, lipoxins) have a variety of biological actions [19]. Nevertheless, their role in inflammation and immunity is the best researched [119,120,121,122,123,124,125] compared to other health aspects. As mentioned before, there are two families of relevant PUFAs: the n-3 and n-6 groups. As a generalisation, the n-6 PUFAs give rise to inflammatory mediators while the n-3 form neutral or anti-inflammatory compounds [6,124,125,126,127]. As discussed later (see Section 7 and Section 8), this has led to interest in the ratio of dietary n-3/n-6 PUFAs with regard to consumer health [6].

As a result of infection, the inherent immunity response activates PLA2, COX-2, and 5-LOX so that there is an increased capacity for eicosanoid production [19]. In many cases, however, the increased production of a particular prostanoid has multiple and, to an extent, conflicting consequences. For example, Calder [19] illustrated these events for PGE2. The latter has pro-inflammatory actions in inducing pain and fever and enhancing vascular permeability to allow white blood cells to infiltrate the site of infection. Nevertheless, PGE2 is also able to reduce the production of pro-inflammatory cytokines as well as 5-LOX (which will inhibit the formation of pro-inflammatory leukotrienes) and induce 15-LOX to increase the synthesis of pro-resolution lipoxins [19,126]. PGE2, therefore, has both pro- and anti-inflammatory actions and is a regulator of inflammation with activity in its initiation, propagation and resolution phases [19]. Moreover, the actions of PGE2 towards different cells (dendritic cells, T lymphocytes, B lymphocytes, T-helper 1 cells, etc.) point to the complexity of responses towards individual eicosanoids.

PGE2 is usually the most abundant PG in tissues [127]. It is released into the intracellular space and rapidly converted to 15-keto PGE2 which is essentially inactive. Non-steroidal anti-inflammatory drugs, including aspirin, cause acute reductions in PGE2 through the inhibition of COX activity [128]. PGE2-EP receptor-mediated effects have notable actions on acute and chronic inflammation as well as for some autoimmune diseases [127]. The mechanisms involved in PGE2-induced several acute inflammatory conditions have been reported e.g., [129,130]. Downstream events often involve IL-6 production and cellular Ca++ changes. PGE2-EP2/EP4 signalling is involved in T cell differentiation [131] with effects on TLR signalling via the cAMP-protein kinase A pathway [132]. PGE2 will also affect Th17 cell expansion [133] and induce IL-23 production from dendritic cells [127] and amplify this expression through canonical and non-canonical NFκB pathways [134]. Some autoimmune diseases (contact hypersensitivity, autoimmune encephalomyelitis, atopic dermatitis, and multiple sclerosis) also appear to be strongly influenced by PGE2-EP2/EP4 signalling [127].

PGD2 is an allergic and inflammatory mediator when released from mast cells [120,122]. It is important for the induction of asthma and can have pro-inflammatory actions on the skin to cause erythema, oedema, and leukocyte infiltration [135]. PGD2 also has multiple pro-inflammatory effects in several animal models [19]. PGD2 is one of the eicosanoids (along with PGA1, PGA2, PGD1 and PGJ2) which activates PPAR [136] and this activation is not found with many prostaglandins. Using a competition binding assay, the eicosanoids 8(*S*)-hydroxyeicosatetraenoic acid and 15-deoxy-PGJ2 (15d-PGJ2) bound to PPAR alpha and PPAR gamma, respectively, at physiological concentrations [137]. Further work on 15d-PGJ2 as a natural PPARgamma agonist has revealed that it can play many roles in anti-tumour and anti-inflammatory processes [138]. Simultaneously, PPARs can influence NFkappaB and JAK-STAT which regulate the expression of downstream genes and mediate inflammatory diseases such as arthritis [139,140] as well as cancer [141]. Moreover, the connection between PPARgamma with 15dPGJ2 activation and different cancer lines has been confirmed [142]. A similar connection has been reported for reductions in inflammatory parameters [143].

LOX products are also important in inflammation [19,120,122,144]. LTB4 is a potent chemoattractant and acts to recruit leukocytes to inflammatory sites. It can also act to increase the adhesion of these cells to the endothelium. Other pro-inflammatory effects include increasing the production of reactive oxygen species and pro-inflammatory cytokines [19]. The cysteinyl-LTs (LTC4, LTD4, LTE4) are produced by a number of cells (e.g., macrophages) following stimuli such as allergens. This, in turn, can have a role in asthma as well as other pro-inflammatory responses [119,145,146].

While 5-LOX produces various leukotrienes, other LOXs give rise to different lipid mediators [18,19]. 15-LOX (15-LOX type 2) metabolises ARA to 15-HpTE which can be reduced to 15-HETE. The latter will form 5,6-epoxyeicosatetraenoic acid (by 5-LOX activity), a precursor of the lipoxins LXA4 and LXB4 [147]. Epimers of LXA4 and LXB4 can also be produced by COX-2 activity in the presence of aspirin and are termed aspirin-triggered LXs [147]. In contrast to the leukotrienes, LXs are SPMs and, therefore, anti-inflammatory.

The divergent functions of pro- and anti-inflammatory products of 5-LOX have been discussed in detail [148] as well as possibilities for new targets to inhibit. For example, antagonists of LTC4 or LTD4 can compete for binding to their G-protein-coupled receptor [148] while inhibitors of LTA4 hydrolase [149] or LTC4 synthase [150] have been developed. In contrast to the leukotrienes, lipoxin binds to different receptors (e.g., FPR2/ALX) and interferes with LTB4 functions and antagonises cysteinyl-leukotriene receptors 1 and 2 to block pro-inflammatory responses [148]. Thus, lipoxins reduce vascular permeability, stop excessive neutrophil infiltration and activation, and are chemoattractants of monocytes for wound healing [147]. The balance between LTs and SPMs such as lipoxins in leukocytes might be influenced significantly by differential intracellular localisation of 5-LOX [151]. Moreover, experiments with macrophages using a FLAP inhibitor which inhibited 5-LOX production of LTs while allowing SPM production emphasized the molecular details of the production of pro- and anti-inflammatory LOX products [152]. While LTs are known to impact several important diseases (arthritis, cardiovascular disease, cancer, neuropathology in Alzheimer’s disease) lipoxins can alleviate oxidative stress connected to inflammation [51]. Reactive oxygen species (ROS) activate the NFκB pathway which causes levels of inflammatory cytokines (IL-6, TNFα etc.) to rise. These cytokines sensitise PLA2 to release arachidonic acid which can then be converted into lipoxins. Lipoxins can then reduce the production of ROS [153]. Lipoxin inhibits activation of the NFκB signalling pathway and is of benefit in intracerebral haemorrhage [51], bacterial pneumonia [154] and osteoarthritis [155]. In addition, some of the anti-inflammatory actions of lipoxin can be through reducing NO generation [156]. In fact, lipoxin could inhibit the production of both ROS and NO [51]. Indeed, ATLX4 could have an important anti-inflammatory action through NO [157].

The aspirin-triggered lipoxin A4 (in addition to lipoxin) can be considered a breaking signal in inflammation [156]. The effects of ATL on LPS-induced NFκB activation, phosphorylation of MAPKs and activation of activator protein-1 have been studied. The results showed that ATL inhibited NO and pro-inflammatory cytokine production at least in part via NFκB and p38MAPK signalling pathways. This has implications for neurodegenerative diseases [156]. In addition, the receptors for lipoxins (ALX/FPR2) and their interactions with lipoxin A4 are important for the chronic inflammation associated with cardiovascular disease [107,156,157].

The regulation of 5-LOX is described in detail in [158]. These authors detail the regulatory domain of 5-LOX with phosphorylation sites and binding positions for Ca, diacylglycerol etc. The action of dicer (a multi-domain RNA helicase) is notable. 5-LOX is an effector and regulator of p53 and, as such, is important in cell/tissue growth [159]. This can have implications for cancer and 5-LOX can also be important as a regulator of Wnt and in other developmental pathways [160].

Recently, the characteristics and functions of 12/15-LOX have been described in detail [38]. This enzyme can have both pro- and anti-inflammatory actions. For example, the metabolite 12(*S*)-HETE is a strongly inflammatory attractant for neutrophils and leukocytes in heart tissue [161] while, in the brain, 12(S)HETE and 15(S)HETE both have anti-inflammatory actions [162]. The pro-inflammatory actions of 12(S)HETE and 15(S)HETE have been reported for several tissues [38]. Sometimes these effects involve the induction of pro-inflammatory cytokines. This may be caused by changes in 12(S)HETE such as in its effects on LPS-induced pulmonary inflammation [38]. Conversely, 12/15-LOX regulates the expression of pro-inflammatory eosins in epithelial airway cells [163]. Moreover, increased expression of some of these inflammatory molecules was also dependent on nuclear factor (NFκB) activation [38]. When increased 12/15-LOX expression is found in asthma patients, its product 15(S)-HETE correlates with disease severity [38].

The anti-inflammatory actions of 12(*S*)HETE and 15(*S*)HETE in ischemic brains were reported to be dependent on PPARγ [162,164]. Moreover, in arthritic mouse models, deletion of 12/15-LOX increased joint disruption [165]. In addition, because 12/15-LOX is involved in the production of SPMs, such as protectin D1, the resolution of inflammation can be impaired significantly [38]. In addition, lipoxins generated from 15(*S*)HPETE are also pro-resolving and this benefit can overcome the pro-inflammatory action of HETEs [166]. Lipoxin A4 has several beneficial effects on inflammation, including reducing neutrophil recruitment and clearance of apoptotic cells by macrophages [38]. Such functions appear to be mediated via the G-protein-coupled receptor ALX [167]. Such benefits of SPMs, produced via 12/15-LOX activity, have been reported in a variety of situations including chronic air inflammation, T cell migration and apoptosis, epithelial wound healing and bacterial-mediated peritonitis [38]. Thus, it is obvious that 12/15-LOX and its metabolites can have both pro- and anti-inflammatory actions. These are summarised in Table 2.

Another subject is intimately associated with inflammation. That is, clearance of apoptotic cells is important for the resolution of infection and its related inflammation. Phagocytosis of apoptotic cells causes the non-immunogenic removal of damaged cells as well as autoantigens [168]. It is noteworthy that resident macrophages expressing 12/15-LOX do the primary clearance whereas other phagocytic cells (which do not express 12/15-LOX) rarely participate. This process may be improved by 12/15-LOX metabolites such as lipoxins [38].

Although 12/15-LOX can produce both pro- and anti-inflammatory metabolites (Table 2), their nature is influenced by the substrates they react with. As noted before, in general, metabolites from n-3 PUFAs, such as DHA, are significantly anti-inflammatory. Indeed, 17-HDHA produced from DHA is a more potent agonist against PPARγ than DHA itself [38].

For a detailed survey of 12/15-LOX potential actions in relation to cardiovascular disease, diabetes, neurological diseases and obesity, the reader is directed to the review by Singh and Rao [38].

Arachidonate 15-lipoxygenase type B (ALOX15B) is one of two lipoxygenases that catalyse peroxidation at the 15-position. Details of the reaction mechanism have been described [169,170,171]. It produces 15-HpETE from arachidonic acid [169] but its main product is 15-HETE. In addition, it will oxygenate EPA, DHA, and linoleic and γ-linolenic acids. Moreover, it has a special characteristic in that it can oxygenate ester-lipid-bound PUFAs as well as lipoproteins [49,172,173]. 15-LOX orthologs have been implicated in the synthesis of pro-resolving lipoxins [174]. ALOX15B has important functions in atherosclerosis, inflammatory lung and skin diseases, arthritis, cancer, and several other complaints [169].

Products of cytochrome P450 eicosanoids can be important for inflammation. As mentioned before, the cytochrome P450 oxidases (CYPs) are a super-family of enzymes which can generate lipid mediators in the HETE and EET families from arachidonic acid. Various CYPs can also oxidise the n-3 PUFAs, EPA and DHA. In fact, some CYPs show a selectivity towards n-3 PUFAs [95] with epoxide products being particularly important. In addition, various oxylipins can be produced from linoleic acid. They may activate NFκB pathways and cause oxidative stress [64] or affect obesity through PPARγ [175,176]. The HETEs are regarded generally as inflammatory. Thus, 5-HETE can cause airway constriction, 12-HETE enhances platelet aggregation, 15-HETE produces vasoconstriction, and 20-HETE constricts vascular smooth muscle [62]. In contrast, some HETEs like 5-HETE have been noted to have anti-inflammatory actions such as regulating anti-oxidant responses [177].

The connection of HETE products in relation to inflammatory-related diseases has been reviewed by Ni and Liu [178]. In particular, they note the effects of 20-HETE on the cardiovascular system [179], kidney [180], brain [181], and lung function [179,181]. 19S-HETE was also noted to be of benefit to cardiovascular function [182].

The EETs work through several different mechanisms and are regarded generally as anti-inflammatory [62]. They may contribute to lowering blood pressure through the manipulation of sodium levels [183]. 5,6-EET inhibits B-cell proliferation, possibly through inhibition of NFκB, while 8,9-EET and 11,12-EET have beneficial effects on pulmonary artery endothelial cell proliferation [184]. 14,15-EET suppresses mitochondrial apoptosis [184]. Physiological concentrations of EETs decreased cell adhesion molecule expression and prevented leukocyte adhesion to the vascular wall by a mechanism involving NFκB inhibition [62].

CYP products from EPA have a variety of beneficial effects on blood pressure, neuronal function, steatohepatitis, adiposity and lung cancer [62]. Products from DHA were often vasodilators and protected against oxidative stress and vascular dysfunction [185].

Early work on the cytochrome P450 eicosanoids showed that several (including 14,15-DHET, 11,12-EET, and 20-HETE) were potential activators of PPARγ. In addition, these eicosanoids could enhance CYP4A1, sEH, and CYP2C11 expression, suggesting that they could regulate their own levels [69]. Recently, CYP450 metabolites, such as 14,15-DHET, have been shown to activate PPARγ, which in turn inhibited NFκB-dependent bacterial clearance during post-influenza superinfection [186]. This was unexpected because PPARγ has been shown in several infection and disease models to exert anti-inflammatory activity [187]. The authors suggest that the complexity of the superinfection could result in an aberrant PPAR activation [186]. Nevertheless, there is a body of evidence that eicosanoids derived from CYP metabolism of arachidonic acid have potential as novel preventative and therapeutic targets for inflammatory diseases including those such as cardiovascular complaints [39] that have chronic inflammation as a contributive factor.

### 6.2. Roles of Specialised Pro-Resolving Mediators in Inflammation

As mentioned previously, uncontrolled inflammation is a significant factor in a host of important diseases, including cardiovascular complaints, arthritis, cancer, metabolic syndrome, neurological diseases, chronic pain, as well as bacterial or viral infections. Although acute inflammation is usually protective and self-limiting, if it persists then it is harmful. This has led to increasing awareness of the dangers of chronic inflammation and the proposal that SPMs may have considerable potential for clinical treatments. Two recent reviews summarise the proposed metabolism, function, and utilisation in disease for the different SPMs [11,22].

### 6.3. SPM Production in Diseases and on Challenge

While there are numerous clinical studies emphasising the role of dietary PUFAs in health and diseases, there are an increasing number of reports about the specific involvement in n-3 PUFA-derived SPMs. Thus, dietary supplementation with n-3 PUFA or marine oils should increase SPM synthesis to correlate with better phagocyte function [11]. In contrast, SPM biosynthesis is reduced in several diseases such as multiple sclerosis, osteoarthritis, and tuberculous meningitis [8]. In addition, it has been noted that SPM biosynthesis in COVID-19 patients is lowered (as revealed in bronchoalveolar lavages, plasma, and serum samples) [22].

The proposed routes for the formation of the SPMs (protectins, resolvins, maresins) and their peptide-conjugates have been covered well in recent reviews [11,22] (see Section 3). More details about the molecular details of their enzymatic biosynthesis will be found in [21,89,95,188].

In an experimental system for bacterial-triggered skin inflammation, SPMs are produced locally at the site of infection. These include several resolvins as well as lipoxin LXB4 which (when applied at physiological concentrations) reduced PMN numbers [189]. In the same model system, oral treatment with a cannabinoid anabasum increased resolvins, reduced pro-inflammatory leukotriene LTB4, and promoted resolution [190]. Moreover, in LPS-challenge experiments, the role of SPMs in inflammation resolution was confirmed [11]. Macrophages, which are involved in inflammatory reactions, can be divided into M1 and M2 types. M1-macrophages respond to bacterial infections by producing pro-inflammatory mediators such as LTB4 and PGE2. In contrast, M2-macrophages are proposed to produce SPMs. Moreover, a dual inhibitor of PGE2-synthase 1 and 5-LOX selectively reduced LT and PG production in M1-macrophages while increasing SPM (resolvins, maresins) production in M2-macrophages [191].

#### 6.3.1. Production of Resolvins

The potential functions of the various types of SPMs have been described well in recent reviews [11,19,22]. The E-series resolvins are formed from EPA and RvE1 protects tissues from leukocyte-mediated injury as well as excessive pro-inflammatory responses [12]. This resolvin regulates vascular inflammation by protecting against stenosis including modifying oxidised LDL uptake [11]. RvE1 reduces neuroinflammation in an Alzheimer’s disease model and, in addition, shows benefits against cancerous cells [192]. Other E-series resolvins exhibit similar benefits against harmful inflammatory situations. RvE2 prevents PMN infiltration in peritonitis, RvE3 acts on airway allergic inflammation, and RvE4 has several actions in physiological hypoxia [11].

Resolvin E1 reduces injury-related vascular neointima formation by inhibiting inflammatory responses [193] and early treatment with this resolvin facilitates myocardial recovery from ischaemia in mice [194]. In fact, some studies have shown that RvE1 and other SPMs are reduced in patients with atherosclerosis [195]. In addition, RvE1 can regulate the functions of macrophages by stimulating efferocytosis in murine models [196]. Other animal models have been used to show that RvE1 can decrease atherogenesis [197] and reduce the size of atherosclerotic lesions [198]. The potential beneficial roles of resolvins and other SPMs in cardiovascular disease have been reviewed [8,107].

The D-series resolvins are produced from DHA either from LOX action (RvDs) or by COX-2 in the presence of aspirin (AT-RvDs/17R-RvDs) [11,18]. Both RvD1 and 17R-RvD1 are reported to regulate phagocytosis at very low concentrations. Thus, they may have benefits against inflammation in Parkinson’s disease, bacterial-induced lung inflammation, and tumour growth [11,199]. These resolvins may also benefit ischemia following tissue injury such as in ischemia/reperfusion-induced kidney injury [11] as well as regulating leukocyte functions [200]. RvD2 also suppresses tumour growth and may be a potent regulator of bacterial sepsis. It can reduce tissue necrosis after burn wounds and stimulate repair while enhancing muscle regeneration [11]. RvD3 and 17R-RvD3 are reported to show anti-cancer activity and regulate leukocyte actions, while RvD4 has been suggested to have potential for controlling thrombo-inflammatory disease. RvD5 may help control bacterial infections [11] and, interestingly, shows gender dimorphism for pain regulation [201].

While the FPR2/ALX receptor is involved with lipoxin signalling, AT-RvD1 is also an antagonist [107]. However, RvD1 and RvD2 additionally signal via DRV1/GPR32 [107]. As expected, GPR32 is expressed in vascular endothelial [202] and smooth muscle cells [203]. The beneficial effects of RvD1 on endothelial cell integrity and barrier function are blocked by antibodies against DRV1/GPR32 or FPR2/ALX, suggesting that RvD1 acts through both [202]. For RvD2, the receptor is termed DRV2/GPR18 [204,205]. Several other ligands can activate this receptor and, depending on the cell type and stimuli, the intracellular signal is an increase or a reduction in cAMP production [107]. Downstream effects include an increased capacity of macrophages to phagocytose debris and dead cells and a reduction in PMN filtration [204]. In separate studies, RVD1 has been shown to mitigate muscle cell proliferation and leukocyte infiltration [206]. Both RvD1 and RvD2 can inhibit VSMC and monocyte proliferation, migration and adhesion [207] while a new RvD1 mimetic served as a potent immunoresolvent [208].

Further beneficial effects of RvD1, AT-RvD1 and RvD2 are summarized in [188]. These include diminishing neutrophil infiltration [209,210,211], reducing the NFκB signalling pathway [212], and promoting resolution and survival in infectious inflammation [213,214,215]. General reviews are [8,188].

#### 6.3.2. Biosynthesis of Protectins and Maresins

A second group of SPMs are the protectins synthesised from DHA via 15-LOX action [11,18]. Although protectins were originally researched in brain tissue, following aspirin treatment (and termed neuroprotectins), they are found in many tissues [18]. They have been reported to have acute actions in many disease states [11]. For example, in brain tissues where DHA is a major fatty acid [216,217], NPD1/PD1 can protect against ischemic stroke, retina degenerative disease [218], and traumatic brain injury [11]. The same protectin seems to be involved in asthma and with macrophage activity connected to inflammatory pain. PD1 is protective in many models of infection and sepsis [22]. A positional isomer of PD1 can be generated from DHA by the action of two plant LOXs. It is now called PDX and differs from PD1 in the geometry of double bonds in the conjugated triene. PDX inhibits platelet activation and improves insulin sensitivity and atherosclerosis in type-2 diabetes [219]. Both PD1 and PDX suppress the replication of the influenza virus [101]. In addition, the aspirin-triggered epimer 17*R*-NPD1 has similar actions to NPD1 [11].

A recent update about the biosynthesis of protectins has been made [89]. This goes into detail about the mechanisms of the reactions resulting in the production of protectins (including PDX) and their sulfido-conjugates. Examples of typical pro-resolving mechanisms of protectins (mainly PD1) are given. These include upregulating efferocytosis and phagocytosis, enhancing bacterial clearance and neuroprotection, downregulating NFκB signalling and pro-inflammatory cytokine production, and regulating T-cell migration. These effects have important implications for infection, obesity, cardiovascular disease, neuroinflammation, multiple sclerosis, and sepsis [89].

Confirmation of the general benefits of PD1 is documented in [100] for phagocytosis regulation, protection against inflammation [100], epithelial injury repair [220], and artery conditions [207]. Similarly, protectins have been reported to have protective actions in general for atherosclerosis [8] and bacterial infections [188].

Maresins, a third group of SPMs derived from DHA, are first biosynthesised by 12-LOX activity [11]. Mar1 was first identified in human macrophages. The name derives from macrophage mediators in resolving inflammation. MaR1 has been reported to be pro-repair, pro-regenerative, and neuroprotective in a range of tissues and phyla [221]. It activates a specific G-protein-coupled receptor, LGR6, which is involved in the pro-resolving functions of phagocytes [22]. Another receptor, ROR-α, can be involved in MaR1 actions on liver macrophages. MaR1 can improve recovery after spinal cord injury, reduce inflammation in perioperative disorders and attenuate pain sensitivity [11]. It will also accelerate wound healing and bone regeneration while reducing pain after tooth extraction [101]. Psoriasis-like inflammation and UVB irradiation damage are also reduced in the skin [11].

A variety of beneficial actions of maresins with regard to inflammation have been reported. Some of these relate to the pathogenesis of atherosclosis [8] as well as bacterial infections [188]. MaR1 attenuates inflammatory signalling pathways in smooth muscle and endothelial cells [222]. It also prevents atheroprogression in mice [223], promotes phagocyte immunoresolvent functions [101], and has general PSM actions [224]. With bacterial infections, MaR1 mitigates the inflammatory response and protects from sepsis [225,226]. It will activate LGR6 and thus promote phagocytic immune-relevant functions [101]. For LPS-induced acute lung injury, MaR1 mitigates the process [227] and accelerates resolution partly by preventing neutrophil survival [228].

The peptide-lipid conjugates (cys-SPMs) can have pro-repair and pro-regenerative actions [11]. The maresin conjugates (MCTRs) prevent LTD4-induced airway contraction and alleviate allergic airway inflammation. Together with the protectin conjugate PCTR1, MCTR1 can also prevent adult respiratory distress syndrome and its associated organ failure [229,230].

#### 6.3.3. Further Remarks about SPM Production and Potential Activities

Apart from EPA and DHA, n-3 docosapentaenoic acid (n-3DPA) can also be converted into SPMs. These include resolvins, protectins and maresins as well as 13-series resolvins [11,22]. A number of studies suggest a role for DPA in inflammation [231]. The effects are due to the conversion of DPA into bioactive mediators by similar pathways to those for EPA and DHA. Diurnal regulation of RvDn-3DPA, as well as its addition to the blood of healthy volunteers, support its protective role in cardiovascular disease [22]. In addition, the benefits of RvD5n-3DPA were noted in arthritic inflammation. The same mediator can help during experimental sepsis [22]. 13-series resolvins, which are also formed from n-3 DPA, ensure neutrophil movement to infection sites, increase the uptake and killing of bacteria by phagocytes, and promote the uptake of apoptotic cells by macrophages. Moreover, the production of such mediators is upregulated by statins, which have been suggested to increase their efficacy in reducing inflammation [232,233].

In this section, I have concentrated on the actions of lipid mediators in inflammation *per se*. However, as noted before, many important diseases have chronic inflammation as a causative or pre-disposing factor. Thus, the classic eicosanoids and newly discovered SPMs, as well as elovanoids and isoprostanes, can be important in these diseases. For more details on the actions of lipid mediators in the health or diseases of the cardiovascular system, renal or gastrointestinal function, reproduction, arthritis, or cancer, the reader is referred to recent publications [11,12,18,19,22,38,148]. The role of SPMs in glial cell function, and hence in nervous system development, homeostasis, and response to injury, have been reviewed recently [234]. Receptors for all the main SPMs (including LXA4) have been noted with differences in their effects on astrocytes and neurons. As in other tissues, the binding of LXA4 to the ALX/FPR2 receptor inhibits the expression of pro-inflammatory cytokines via an NFκB-dependent mechanism [235]. SPMs strongly reduce microglial activation and their production of pro-inflammatory mediators, and increase their ability to phagocytose and remove aggregated proteins such as amyloidβ [236,237]. These actions have relevance to several models of neurodegenerative conditions such as Alzheimer’s disease, multiple sclerosis, and Parkinson’s disease [234]. In addition, Ponce et al. [238] reviewed the role of SPMs in reducing neuroinflammation associated with neurodegenerative disorders. Alzheimer’s and Parkinson’s diseases were particularly emphasised, along with the links between pro-inflammatory compounds and signalling pathways such as NFκB. Notably, the beneficial effects of NPD1 on beta-amyloid precursor protein metabolism were noted [239]. Such observations suggest that SPMs could be useful as add-on or standalone agents for the treatment of patients with neurodegenerative diseases [238].

Despite an impressive number of publications from different laboratories documenting the presence and biological activities of various SPMs, some key aspects have been questioned recently [81]. These authors challenge the analytical methodology when applied to leukocytes and other cells. They suggest that concentrations of SPMs in vivo are too low to have the biological effects claimed. Moreover, SPMs in tissues are too low and too unstable to be assayed with any confidence. Furthermore, SPMs did not increase with supplements of their omega-3 precursors and the evidence for their appearance during the resolution phase of inflammation was inadequate. In addition, the identity of G-protein-coupled receptors needs to be confirmed by independent means.

What is not in dispute seems to be the detailed structural analysis and synthesis work to identify the various SPMs that can be formed from EPA and DHA. Moreover, the increasing body of evidence showing that SPMs have beneficial effects when applied *at sufficient concentrations* in various disease states is important and suggests significant future clinical applications.

A meeting (since the publication of [81]) in Sweden, where the different parties were present, failed to resolve the points at issue and future clarification is needed. In the meantime, it seems prudent for journals (and their referees) to be fully cognizant of the questions raised about studies of SPMs and the difficulty of robust analysis of these unstable molecules in biological samples.

## 7. Production of VLCPUFAs from Dietary Essential Fatty Acids

As commented previously, the dietary EFAs, first elucidated by the Burrs, are the typical higher plant PUFAs, linoleic and α-linolenic acids [18]. However, the various lipid mediators are produced from VLCPUFAs which are 20C or 22C in length and contain additional double bonds. In humans, they are produced from linoleic and α-linolenic acids by a series of desaturation and elongation reactions as illustrated in Figure 4. Almost all of the reactions needed take place on the endoplasmic reticulum and usually begin with a Δ-6 desaturation. There is an alternative pathway using an elongation first (Figure 4). The supply of the EFAs is due to the fact that oxygen-evolving photosynthetic organisms (cyanobacteria, algae, lower and higher plants) contain Δ-12 and Δ-15 desaturases. It is recognised that dietary EFAs are converted to VLCPUFAs (especially into DHA) rather poorly in mammals [5,217] so there has been considerable interest in finding sources of EPA and DHA for the diet. The vast majority of VCPUFAs are produced by algae which lie at the base of food chains [240]. However, because desaturation reactions tend to be reduced at higher temperatures, the supply is under threat due the climate change [6,241]. In fact, this whole topic has been reviewed in detail recently [242]. Moreover, because of the poor conversion of EFAs into VLCPUFAs in mammals, a dietary supply of EPA and DHA can be considered ‘conditionally essential’ in certain circumstances [5] and may also be considered advisable for certain individuals, especially newborns [243,244,245,246].

Although conversion of EFAs into VLCPUFAs is needed for dietary PUFAs to be effective, the speed of production of EPA and DHA has been questioned and carefully re-examined recently [247]. The authors make several important conclusions. First, they consider that the conversion of ALA into EPA and DHA may be higher than previously thought. Second, there is an unsuspected complex relationship between tetracosahexaenoic acid (24:6 n-3) and DHA and, third, the increase in EPA levels on feeding DHA is only partly due to retroconversion [247]. In addition, it should be noted that ALA is essential in the diet even when VLCPUFAs are present. This may be due to the different utilisation of dietary DHA and liver-derived DHA. Thus, these authors conclude that dietary ALA is critical in maintaining tissue levels of n-3 PUFAs [248].

One feature of the conversion of EFAs into VLCPUFAs as illustrated in Figure 4, is the so-called ‘Sprecher pathway’ involving a 24:6 intermediate followed by β-oxidation. This is named after its discoverer who could not detect a Δ4-desaturase in mammalian tissues to convert 22:5 into DHA (Figure 4) [249]. The Sprecher pathway is key in humans and other mammals and, with the exception of a few teleost species, in fish also [250].

### Sources of VLCPUFAs for Humans

In human diets, fish and fish oils are the main sources of VLCPUFAs. Fish should normally have an adequate supply of such acids in their feeds, whether they are from capture fishing or from aquaculture. In fact, aquaculture now supplies more than half the total fish used [251] and about 75% of marine-sourced fish oils are used in aquaculture [252]. Because this situation is not sustainable where demand for VLCPUFAs (particularly EPA and DHA) is rising [253], there is much importance attached to how aquaculture can be adapted for the future. In particular, because so much fish oil is currently used in aquaculture and wild fish catches are dwindling, efforts are being made to substitute plant oils in the feed [254,255].

It is important to point out that the enzymes used for the production of VLCPUFAs in mammals (Figure 4) have activity with both n-3 and n-6 PUFAs. Consequently, the ratio of such acids in the diet is important. Moreover, ‘Western’ diets usually have a ratio of linoleic/α-linolenic acids of 10 or more whereas a ratio of 3–4 is recommended for good health [6,217,243]. This aspect needs urgent attention in order to provide suitable recommendations for everyday intake of dietary fats.

Since fish are the main dietary sources of VLCPUFAs (especially n-3 PUFAs) and there is an increasing shortage of them around the world, alternative sources of such acids have been sought. Crops have been genetically modified to synthesise EPA and/or DHA [254,255,256,257]. Such manipulations have either used a polyketide synthase system [257] or a combination of mainly algal genes to make significant amounts of EPA and/or DHA [258,259]. Currently, there are a number of potential commercial sources for such acids [255,256] (see Section 8).

Although fish are the main source of VLCPUFAs in typical human diets, the content of such acids depends on their feeds. However, there is not a simple relationship between the algae (or fish oil) consumed and the final fish content. Thus, the relationship depends on a multitude of factors, including the environment as well as fish development and feeding strategies (particularly ‘finishing diets’) [260]. It has generally been considered that most fish have a somewhat limited ability to biosynthesise VLCPUFAs. However, there are exceptions such as the rabbit fish (*Siganus canaliculatatus*). Moreover, relevant desaturases or elongases have been found in many species. These findings are of special relevance in view of the widespread use of fish oils in aquaculture [261]. Nevertheless, despite these recent findings, fish (like other vertebrates) lack the Δ-12 and Δ-15 desaturase that provide the EFAs. In contrast, microorganisms such as algae, protists, cyanobacteria and a few bacteria are capable of synthesising EFAs de novo. This is usually done by an aerobic desaturase (and elongase) pathway but can also be achieved by an anaerobic polyketide synthase pathway (such as in Thraustochytrids) [262,263,264]. The microorganisms involved have been well reviewed by Qiu et al. [265].

Marine ecosystems are enriched in n-3 VLCPUFAs where algae are recognised as the main sources [240]. However, recently it has also become clear that many invertebrates (mostly aquatic) contain the necessary enzymes to biosynthesise a variety of PUFAs de novo. For more details see [266] which includes a discussion of VLCPUFA biosynthesis in fish. Actually, the first reports of non-photosynthetic organisms that contain the critical Δ-12 desaturase to make linoleic acid were from the work of Borgeson and Blomquist. They first reported linoleic acid synthesis in two insects, the house cricket (*Acheta domesticus*) and the American cockroach (*Periplaneta americana*) [267,268,269]. Later, they found *de novo* formation of linoleic acid in two non-insect invertebrates, the land slug and the garden snail [270].

## 8. Sources of Dietary PUFAs

Oil crops are the main sources of PUFAs in food and feed. Currently, the main crops are oil palm, soybean, oilseed rape and sunflower, representing about 35%, 26%, 15% and 9%, respectively, of the total consumption [271]. In short, these four crops currently provide 86% of the world’s vegetable oils. Details of the production, composition, processing, and utilisation of major oil crops are given in The Lipid Handbook [272]. The dominant fatty acids of commodity vegetable oils are palmitic, oleic, and linoleic acids. While α-linolenic acid (ALA) is present in most oils, it is usually only in small amounts. Soybean (7%), oilseed rape (6%), and flax (47%) are crops with significant amounts of ALA. However, there is so much linoleic acid in soybean oil that the ratio of n-6/n-3 PUFA is about 8. Only In oilseed rape (Canola varieties) is there a ‘healthy ratio’ (about 2.3) for human consumption [240,271]. As noted before, because the conversion of EFAs into VLCPUFAs (which provide almost all lipid mediators) uses enzymes which work with both n-6 and n-3 PUFAs (Figure 4), their dietary ratio is very important and has led to increased attention being paid to adequate dietary n-3 PUFAs. Dietary effects and recommendations are discussed in [240] while the food applications of lipids are reviewed in [273].

### 8.1. Increasing Oil Crop Yields

There are two main aspects to consider when discussing dietary sources of PUFAs. First, there is a continuous need for more oil year after year [272], the vast bulk of which comes from land-based crops. Second, dietary considerations dictate that the overall composition of oils and their contribution to lipid uses is important [271].

For increasing oil yields in crops, several approaches have been used [271]. These include traditional breeding and agricultural practices. The problems with climate change have increased the importance of both these aspects. For seed oils, quantitative trait loci (QTL) analysis has been used to try and identify useful genes. Moreover, a combination of transcriptome and metabolome analysis can identify regulators that can influence seed oil content [274].

Metabolic control analysis has been utilised to study individual enzymes and quantify their regulatory influence in triacylglycerol formation [275,276]. Such a method allows precise characterisation of how manipulating a particular enzyme can change lipid accumulation. An example was where DGAT1 was up-regulated in *B. napus* and increased the oil content of seeds significantly in both greenhouse [277] and field conditions [278]. Following this example, DGAT is often used in combination with other genes to increase oil accumulation. The synergistic effect of DGAT together with a transcription factor, Wrinkled 1 (WRI1), was shown to channel carbon from carbohydrate metabolism into lipid biosynthesis [279]. In fact, this combination of useful genes to increase oil accumulation has now been taken further with the ‘Push/Pull/Package/Protect’ combination [280]. In this, WRI1 is used to push carbon into lipid synthesis, DGAT pulls the carbon into triacylglycerol formation, oleosin packages oil into droplets, and lipase action is reduced to prevent catabolism. Another example of combining the expression of several genes to augment oil accumulation is given in [281].

The usefulness of flux control analysis in defining the amount of regulation caused by individual enzymes in a pathway has been utilised further. Examination of lysophosphatidate acyltransferase showed that, despite its low intrinsic flux control coefficient, it could contribute significantly to oil accumulation [282]. This agreed with the data of Liu et al. [281], also with *B. napus*. Instead of DGAT, triacylglycerol can also be formed by phospholipid: diacylglycerol acyltransferase (PDAT) in a non-acyl-CoA reaction. When the activity of this enzyme was increased it had a negative effect on oil accumulation and altered lipid metabolism in *B. napus* [283]. This contrasts with the beneficial effect of PDAT in crops producing some unusual oils [283]. Thus, flux control analysis can provide very useful information about triacylglycerol formation in oil crops. Nevertheless, it neglects the integration of lipid metabolism into a genome-scale metabolic network, which has been reviewed recently [284].

### 8.2. New Sources of VLCPUFAs

As mentioned previously [6,240], the main source of VLCPUFAs in human diets is fish where the animals have consumed algae, the actual site of such synthesis [240]. Because this situation is unsustainable, there have been attempts to engineer vegetable crops to produce VLCPUFAs over the last decade, mainly by using algal enzymes. In addition, several commercially useful algae (that accumulate significant quantities of VLCPUFAs) have been studied and ways of increasing their yields as well as their % of VLCPUFAs examined [6,285]. However, unless the algal oils are used for high-value products (such as infant formulae) the cost of growing and processing algae remains a serious problem. Nevertheless, they can be used to provide VLCPUFAs for supplements in eggs, meat, and milk, in pet foods, and especially in aquaculture; see [6,285,286,287,288].

Because of the currently prohibitive cost of producing algal oils, serious efforts have been made to modify plants to produce VLCPUFAs. Examples of commercial genetically-modified plants are listed in Correa et al. [284]. The examples listed are mainly in oilseed rape but two groups have successfully modified *Camelina sativa* to produce EPA and/or DHA contents that are similar to fish oils [289,290]. These research efforts have been driven by the need to develop a land-based source of EPA and DHA to replace dwindling fish oils [254,255]. The most popular approach (as exemplified in the above examples) has been to install an alternative pathway of desaturases and elongases. A major problem with this approach has been the issue of ‘substrate dichotomy’ where desaturation often uses phosphatidylcholine whereas elongation uses acyl-CoA substrates [254,291,292]. The initial attempts with different oilseeds are summarised in [256]. Some of the recent attempts which have seen significant improvements in vegetable oils as important sources of VLCPUFAs are documented in [256,284,293,294]. The present position is that a number of plant crops can provide alternatives to the currently fragile source of VLCPUFAs from fish for use in food or feeds (especially in aquaculture).

## 9. Conclusions

Since their discovery as essential fatty acids, linoleic and α-linolenic acids have been recognised for a number of important biochemical and physiological functions. Their actions are almost always due to their metabolic conversion to lipid mediators. Apart from the classic eicosanoids, recent research has identified many other mediators, such as the SPMs and isoprostanes. Together, the lipid mediators have very important health functions and, moreover, they impact many of the most important human disease states. With this recognition, it has become clear that dietary supplies of the necessary PUFAs, especially VLCPUFAs, are critical. Thus, efforts are being made to replace the current main source, fish oils, with photosynthetic products, especially in land crops. Success in these ventures promises to ensure these vital constituents for future diets.

## Figures and Tables

**Figure 1 ijms-24-08838-f001:**
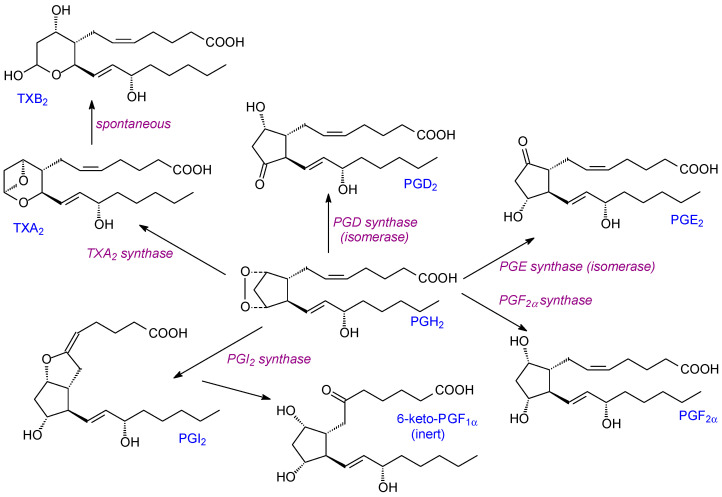
Conversion of cyclooxygenase endoperoxide products to different eicosanoids. The example shown is for PGH2 as formed from arachidonic acid. Arachidonic acid is first converted to the cyclooxygenase endoperoxide product PGH_2_. This, in turn, can form the prostaglandins PGE_2_ or PGF_2_α, prostacyclin PGI_2_ or thromboxanes TXA_2_ and TXB_2_.

**Figure 2 ijms-24-08838-f002:**
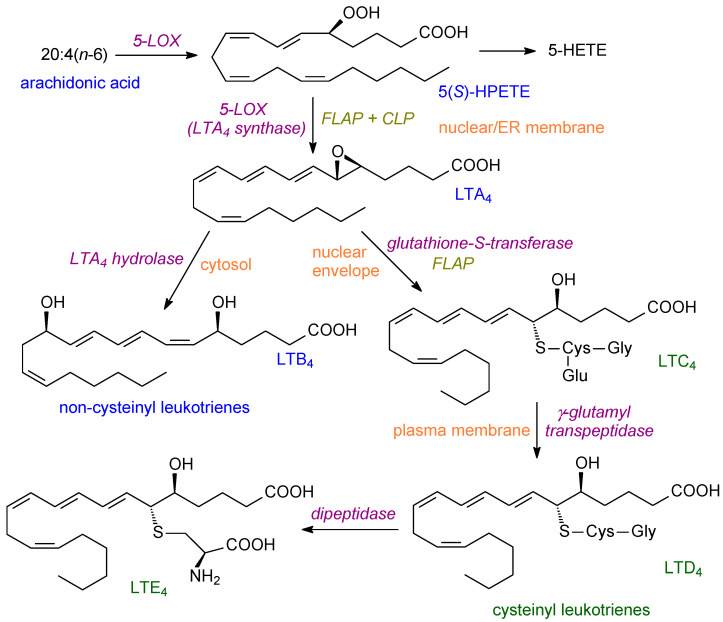
Leukotriene biosynthesis from arachidonic acid. 5-LOX is a key enzyme and catalyses a two-step concerted reaction. In the second step, two accessory proteins, FLAP (5-lipoxygenase activating protein) and CLP (coactosin-like protein) are needed and LTA_4_ is produced. LTA_4_ can either be hydrolysed to form LTB_4_ or converted to various ‘cysteinyl leukotrienes’.Adapted and re-drawn from [18].

**Figure 3 ijms-24-08838-f003:**
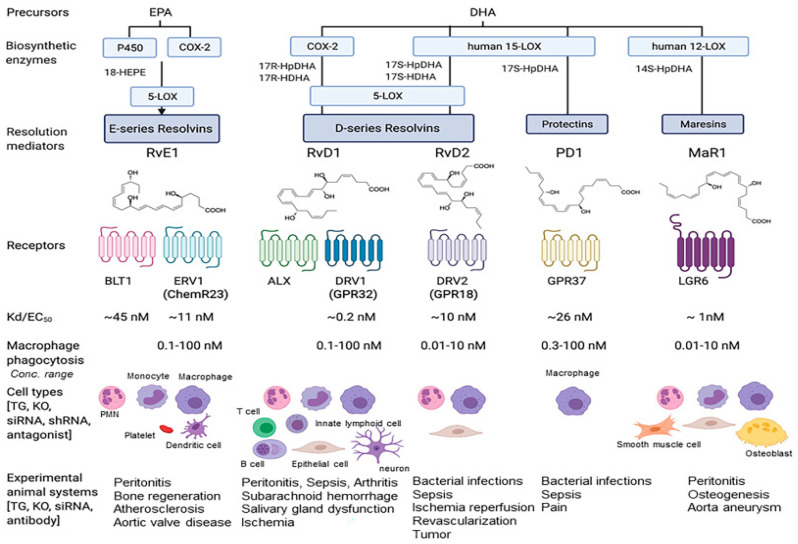
Illustration of the resolution metabolome showing proposed SPM biosynthesis, receptors and functions. Precursors EPA and DHA are converted via biosynthetic enzymes to SPMs, which in turn activate their specific receptors to stimulate pro-resolving innate immune functions. Each SPM demonstrates stereoselective activation of its cognate GPCR on selective cell types, leading to intracellular signals, pathways and pro-resolving functions. The affinities of SPMs for their respective recombination GPCRs (i.e., Kd or EC50 values) are consistent with their bioactive concentration ranges e.g., macrophage phagocytosis (picomolar-low nanomolar) in vitro. The in vivo functions of these SPM receptors were demonstrated using transgenic and/or knock-out mice, as well as specific blockage of the receptor e.g., siRNA, antibodies or receptor agonists (see [22] for details. The figure is taken from [18] under the terms of the Creative Commons License for an Open Access publication. However, please see the end of Section 6.3 for comments arising from the publication of [81] about challenges to some of the proposals for the generation, concentrations in vivo and action of SPMs.

**Figure 4 ijms-24-08838-f004:**
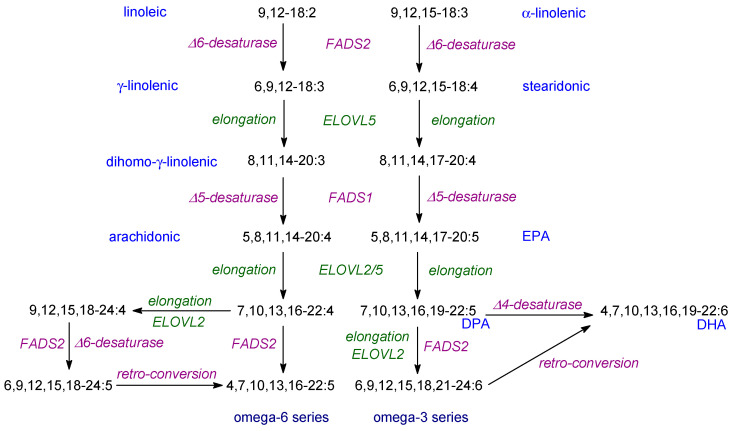
Conversion of the essential fatty acids to 20-and 22-carbon PUFAs. The two EFAs (linoleic and alpha-linolenic acids) are metabolized by the same desaturases (Fads) and elongases (Elovls). Conversion to longer chains usually begins with Δ6-desaturation although an alternative pathway elongates the chain first. In algae, a Δ4-desaturase allows direct production of 22:6 (DHA) but, in animals, DHA is made via a 24C intermediate followed β-oxidation in the Sprecher pathway.

**Table 1 ijms-24-08838-t001:** Complaints where increased n-3 PUFAs have been reported to be beneficial.

Arthritis—rheumatoid, osteoarthritis
Cancer—e.g., colon
Cardiovascular disease
Chronic pain
Crohn’s disease
Neural development—brain development (AHD, cognitive function)
—vision (optic nerve function)
Neurodegeneration—Alzheimer’s disease, dementia
Infections—bacterial, viral (incl. COVID-19)
Ulcerative colitis

**Table 2 ijms-24-08838-t002:** Effects of 12/15-LOX on Inflammation (see [29]).

*Pro-Inflammatory Effects*	
12/15-LOX products	12(S)HETE pro-inflammatory chemoattractant for neutrophils, leukocytes; induces expression of inflammatory cytokines
12/15-LOX activity	Expression of pro-inflammatory eoxins; induces MCP1 expression
	Interacts with LPS to increase inflammation; promotes severity of asthma
12/15-LOX deletion	Reduces the expression of pro-inflammatory genes
*Anti-inflammatory effects*	
12/15-LOX products	Forms lipoxins, protectins, resolvins
12/15-LOXactivity	Pro-resolving effects; regulates dendritic immune function; macrophages with 12/15-LOX activity clear apoptotic cells
12/15-LOX deficiency	Reduced protectin synthesis; resolution of inflammation impaired; wound healing reduced
Deletion of 12/15-LOX	Reduced lipoxin production; increased joint destruction; impairs apoptotic clearance
12/15-LOX expression	Implications for complaints (CVD, diabetes, obesity, neurological)

## Data Availability

Not applicable.

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
