# Peer review of "Polyunsaturated Fatty Acids: Conversion to Lipid Mediators, Roles in Inflammatory Diseases and Dietary Sources"

_ijms, 2023, doi:10.3390/ijms24108838_

Round 1

Reviewer 1 Report (Previous Reviewer 1)

The review “Polyunsaturated Fatty Acids: Conversion to Lipid Mediators, Roles in Inflammatory Diseases and Dietary Sources” is a very interesting work but major information is missing to conclude and there is a major structural problem.

The review is very rich in information on this field and represents a good scientific synthesis

But the author has not invested in the preparation of figures and in the structuring of the tables.

it is not acceptable to take the same figures with the same titles and put them in this review, even worse without mentioning the reference

figure 1 and 2 and 4 are taken from the article Christie, W.W.; Harwood, J.L. Oxidation of polyunsaturated fatty acids to produce lipid mediators. Essays in Biochemistry 2020, 64, 401-421.

figure 3 is taken from the article  Dyall, S.C.; Balas, L.; Bazan, N.G.; Brenna, J.T.; Chiang, N.; da Costa Souza, F.; Dalli, J.; Durand, T.; Galano, J-M.; Lein, P.J.; Serhan, C.N.; Taha, A.Y. Polyunsaturated fatty acids and fatty acid-derived lipid mediators: recent advances in the understanding of their synthesis, structures and functions. Prog. Lipid Res. 2022, 86, 101165.

The author has to valorize the work by preparing figures, especially for this review.  also, the tables need more attention and should be more structured and informative.  

You have to show the major revisions in the text, with a different color text, by highlighting the changes.

All these remarks must be taken into consideration. Answers to these criticisms will reinforce the quality of the manuscript and will permit us to conclude with more accuracy.

Author Response

Please see attachment for the highlighted text.

In the previous resubmission I included a response to reviewers and a covering letter. In both I explained that Figs. 1, 2 and 4 HAD been redrawn and that Fig. 3 had been left because it showed the Serhan group's version of the metabolism and effect of SPMs and I had been careful to remain neutral while the controversy was resolved.

Reviewer 2 Report (Previous Reviewer 2)

The manuscript has been improved and I don't have further questions.

Author Response

Your comments have been acknowledged. Thank you.

Round 2

Reviewer 1 Report (Previous Reviewer 1)

The review “Polyunsaturated Fatty Acids: Conversion to Lipid Mediators, Roles in Inflammatory Diseases and Dietary Sources” is a very interesting work with a good scientific idea.

the author has made a considerable effort in the part of molecular regulation and pathways, which has improved the quality of the review

but the author has to redraw fig 2. because we cannot put the same fig in two articles

figure 2 is taken, without any change, from the article Christie, W.W.; Harwood, J.L. Oxidation of polyunsaturated fatty acids to produce lipid mediators. Essays in Biochemistry 2020, 64, 401-421.  

the author must decrease the number of uses of the reference “Christie, W.W.; Harwood, J.L. Oxidation of polyunsaturated fatty acids to produce lipid mediators. Essays in Biochemistry 2020, 64, 401-42” because it is used more than 20 times in this review !

the article will be accepted after minor revision

Author Response

Figure 2 HAS been re-drawn and is new. Moreover, the citations for the reference concerned were reduced at the first revision of the manuscript, as noted in the comments to reviewers at the time.

This manuscript is a resubmission of an earlier submission. The following is a list of the peer review reports and author responses from that submission.

Round 1

Reviewer 1 Report

This is an interesting work but major information is missing to conclude and there is a major structural problem.

# Abstract:

To have the same logic as the title you have to start the abstract with sources part and finish with roles 

The abstract doesn’t mention the objective of this review.

1. Introduction:

You have to reorganize the table1 and put the references of the effects

In line 55 you have this sentence “Table 1 lists diseases where n-3 PUFAs have been shown to have beneficial effects.”  

but the references that you mention after the table speak about the importance of Uncontrolled and/or chronic inflammation in pathological situations  line 72  “Uncontrolled and/or chronic inflammation is important in many widespread, pathological situations. These include arthritis, cancers, cardiovascular diseases, chronic pain, 73 neurological complaints as well as bacterial and viral (including Covid-19) infections [8,9]” 

they are not the same things!.

2. Conversion of polyunsaturated fatty acids into lipid mediators

You have to create a title “line 86”    2.1 Eicosanoid biosynthesis,  so   2.1. Eicosanoid Receptors will be 2.2

Figure 1: the figure is blurred and you have to work on it to can be more explanatory

Line 128 you have to change this sentence “Other features 128 of the reaction are summarised in [15]”

In this paragraph you start with humans, you speak about mice and animals, and after you return to humans’ genes.

“In humans, six main families of LOXs have been 124 identified----5-LOX, 12-LOX, 12/15-LOX (15-LOX type 1), 15-LOX type 2, 12R-LOX and 125 epidermal LOX [28]. In mice there are seven LOXs. All of the animal LOXs have a single 126 polypeptide chain of 75-80 kDa mass. A beta-barrel at the N end is used for substrate 127 acquisition and a catalytic domain contains a single non-heme iron atom. Other features 128 of the reaction are summarised in [15]. In humans there are six functional genes (ALOX5, 129 ALOX12, ALOX12B, ALOX15, ALOX15B, ALOXE3), four pseudogenes and an ALOX12 130 antisense gene [29].”

You have to start with animals LOX and past to the humans part to have fluidity in the transition to 5-LOX.

You have to remove “These are listed in [16]”. Line 164

3. Specialised Pro-Resolving Mediators (SPMs)

You have to remove    line 175   “This is a controversial area which is discussed at the end of 175 section 6.3. I mention that they 176 have implications for sections 3 and 6”

In this paragraph we have many information without references

Line 180 « Inflammation itself is a natural protective process where the body re- 180 sponses to harmful stimuli such as pathogens or wounding. It functions to clear out dam- 181 aged tissues and necrotic cells as well as eliminating the initial cause. Acute inflammation 182 is actively terminated by its resolution which have been proposed to involve several 183 mechanisms including the production of SPMs as well as the down-regulation of pro-in- 184 flammatory substances (such as leukotrienes). If resolution does not occur, chronic inflam- 185 mation will ensue with severe consequences. »

3.1. Receptors for SPMs

I think is preferable to remove this type of sentence, “Paragraph line 215   As discussed in [15]   Details of the individual reported GPCRs are 221 given in [8,18].  but see also the detailed 226 comments in [39] about the receptors. 227 »  

6.2. Roles of Specialised Pro-Resolving Mediators in Inflammation

this part is underestimated and poor in information despite its importance in your subject, it must be developed and enriched scientifically

6.3. SPM Production in Diseases and on Challenge

This important part must be valorized by a figure

8. Sources of Dietary PUFAs

This part can be lightened with a table

to valorize your work, the figures should be prepared for this review because all the figures you have in the work are already used in previous articles. Also, the tables need more attention and should be more structured and informative.  

The structure of the review needs to be reviewed to have the same logic as the title because the title is “Polyunsaturated Fatty Acids: Sources and Roles in Inflammatory Diseases“  but in the review, you start with the metabolism of the fatty acid and role of the mediators and you finish with the production and sources!

The review needs to be enriched with molecular regulation and pathways in concerning fatty acids oxidation and their roles in diseases and especially in inflammation. (PPAR  and NF-κB)

The review deals with lipids mediators and their roles in inflammation, I think you have to add the mediators to the title.

you have used this reference more than 28 times!   “Christie, W.W.; Harwood, J.L. Oxidation of polyunsaturated fatty acids to produce lipid mediators. Essays in Biochemistry 2020, 64, 401-42” 

All these remarks must be taken into consideration. Answers to these criticisms will reinforce the quality of the manuscript and will permit us to conclude with more accuracy.

Author Response

The title has now been altered.

The objective of the review has now been added to the Abstract.

To clarify this point, Table 1 has been moved above paragraph before. In addition, line 56 of the original text, clarifies the matter.

Sections have now been added to 2, as suggested.

Fig. 1 has been clarified and more explanation has been added to the legend.

Line 128. Additional information has now been added.

The section on LOXs has been reviewed by two independent experts and  considered satisfactory.

For line 164, text has been added to provide more information.

For line 175, because the controversy is so important, I wanted to alert readers at the first mention of SPMs.

Line 180-onwards. References have now been added.

Line 215 onwards. I have tried to keep the text succinct (as recommended by Reviewer 2) so that it did not become too burdensome.

Section 6.2. This has been modified and [18] noted.

Section 6.3. This is addressed in Fig. 3.

8. Sources of dietary PUFAs.  This is a nice suggestion but, unfortunately, the sources are too complex to describe in a Table.

Improvements have been made to the Tables and legends have been added to in the figures.

The new title addresses this point.

The suggestion to add more detail disagrees with Reviewer2.  I have consulted with two more experts and they consider the amount of detail is satisfactory, so I have left the overall amount as it is.

Mediators are now in the amended title

I am sorry for the excessive citation of [15] which was done to save space. The citations have now been reduced in number.

Reviewer 2 Report

1. In my opinion, the topic of this review manuscript is too broad, and it's difficult to focus on in-depth mechanisms regarding the major roles of PUFAs. Maybe the author can narrow down it to just a few diseases and give a comprehensive view of them.

2. In the abstract. the suggestion is to combine the two paragraphs into one. and give key perspectives in the end.

3. The layout of all tables is disorganized and therefore it's not possible to evaluate the manuscript in its current form.

4. All the figures were directly taken from other OA publications and in my opinion, the author should organize new figures by himself and put a comprehensive view into them to enhance the quality of this work. Maybe a book chapter can do this way but a review article in a high-impact journal is not adequate to do so.

5. L14: Delete "they".

6. L15: preventing them becoming chronic - preventing them from becoming chronic

7. L253: An error in the layout.

8. Avoid using first-person writing throughout the manuscript.

9. L633: Don't understand what it means. "We"?

10. The figure legends should be improved significantly to ensure they can be understood when each figure is read separately.

Author Response

  1. This suggestion differs from that of Reviewer 1, who wants more detail. Therefore, I consulted with two more experts who consider that the amount of text is about right. So, I have left the length of the article as it is.
  2.  An additional paragraph has been added to the Abstract to address this point.
  3. The Tables have been reorganised.
  4.  I have used very recent figures from articles by world experts, which I don't think I can improve upon. I have added to the legends to provide a summary view.
  5. Text altered.
  6. Text altered.
  7. Now corrected.
  8. Text corrected.
  9. Text altered to clarify.
  10. The legends have been added to in order to provide more information.

Reviewer 3 Report

This is a useful review by a scientist renowned internationally in the topic, hence it should be really welcome for publication.

I have some comments which aim to make the final paper easier to comprehend by future readers.

One. Possibly sections 3, 4 and 5 may be rearranged between them for easier flow.

Two. Sub-section 6.3 is really lengthy and will be better if divided in two smaller parts.

Three. Same for section 7.

Four. Same for section 8.

Five. The tables are badly formatted and need to be laid in accord with the publishers’ template.

Six. The figures are not as good as they should, there is room for improvement in there.

Seven. Some recent relevant references may also be included.

All in all, the manuscript is heading for acceptance after modification as indicated.

Author Response

  1. I consulted with two other experts about this point and they consider that the order is o.k. Since it follows the arrangement in other recent reviews, I have left it.
  2. Section 6.3 has now been divided.
  3. Again section 7 has been divided.
  4. Also, section 8 has been divided.
  5. The tables have been reformatted.
  6. The figures have been downloaded again to improve resolution.
  7. A significant number of 2022 papers have been included already. I am not aware of any special ones I have missed but would, of course, be willing to include others.

Round 2

Reviewer 1 Report

Unfortunately, the author has not made the requested efforts to improve this version of the review “Polyunsaturated Fatty Acids: Conversion to Lipid Mediators, Roles in Inflammatory Diseases and Dietary Sources” and has not taken the comments into consideration.

The author has made no effort to prepare the original figures for this review (he still uses figures already published in another article) and he has not made the requested development of the scientific parts.

the author only responded to formal remarks and neglected substantive scientific remarks that could enhance the quality of the manuscript and allow us to conclude with more accuracy.

all these reasons led me to reject the manuscript

Reviewer 2 Report

Most of the critical points have not been improved, so I have to reject the manuscript again. Importantly, a reputed journal should publish a paper with original comprehensive graphs but not copy from others. Also, a "molecular" journal should accept a paper that has comprehensive views on molecular mechanisms.

Reviewer 3 Report

The manuscript has been improved and is almost ready for acceptance, subject to a final polishing to correct various linguistic slips throughout the text.